# Dynamic Evaluation of Vitamin D Metabolism in Post-Bariatric Patients

**DOI:** 10.3390/jcm13010007

**Published:** 2023-12-19

**Authors:** Alexandra Povaliaeva, Artem Zhukov, Alina Tomilova, Axenia Bondarenko, Maksim Ovcharov, Mariya Antsupova, Vitaliy Ioutsi, Ekaterina Shestakova, Marina Shestakova, Ekaterina Pigarova, Liudmila Rozhinskaya, Natalia Mokrysheva

**Affiliations:** The National Medical Research Centre for Endocrinology, 117292 Moscow, Russia; a.petrushkina@yandex.ru (A.P.); gavrilova.alina@endocrincentr.ru (A.T.); axenia.bondarenko@gmail.com (A.B.); ovcharov.maksim@endocrincentr.ru (M.O.); antsupova.marya@endocrincentr.ru (M.A.); ioutsi.vitalij@endocrincentr.ru (V.I.); shestakova.ekaterina@endocrincentr.ru (E.S.); shestakova.marina@endocrincentr.ru (M.S.); pigarova.ekaterina@endocrincentr.ru (E.P.); rozhinskaya.ludmila@endocrincentr.ru (L.R.); mokrisheva.natalia@endocrincentr.ru (N.M.)

**Keywords:** vitamin D, vitamin D deficiency, bariatric surgery, obesity, mass spectrometry

## Abstract

Background: findings from the previously conducted studies indicate altered regulatory mechanisms of calcium and vitamin D metabolism in obese patients and a role for bariatric surgery in regulating vitamin D metabolism; however, the available data is controversial and does not provide an adequate understanding of the subject. Methods: we evaluated serum parameters of vitamin D and mineral metabolism (vitamin D metabolites (25(OH)D_3_, 25(OH)D_2_, 1,25(OH)_2_D_3_, 3-epi-25(OH)D_3_, and 24,25(OH)_2_D_3_), vitamin D-binding protein (DBP), free 25(OH)D, fibroblast growth factor 23 (FGF-23), parathyroid hormone (PTH), total calcium, albumin, phosphorus, creatinine, magnesium) in 30 patients referred for bariatric surgery in comparison with 30 healthy volunteers of similar age, sex and baseline 25(OH)D_3_. Patients were also followed up with repeated laboratory assessments 3 months and 6 months after surgery. During the first 3 months, patients were prescribed high-dose cholecalciferol therapy (50,000 IU per week), with subsequent correction based on the results of the 3-month visit examination. Results: Preoperatively, patients with morbid obesity were characterized by a high prevalence of vitamin D deficiency (median 25(OH)D_3_ level 11.9 (6.8; 22.2) ng/mL), significantly lower levels of active vitamin D metabolite 1,25(OH)_2_D_3_ (20 (10; 37) vs. 39 (33; 50) pg/mL, *p* < 0.001), lower serum albumin-adjusted calcium levels (2.24 (2.20; 2.32) vs. 2.31 (2.25; 2.35) mmol/L, *p* = 0.009) and magnesium levels (0.79 (0.72; 0.82) vs. 0.82 (0.78; 0.85) mmol/L, *p* = 0.043) with simultaneous similar PTH levels (*p* = 0.912), and higher DBP levels (328 (288; 401) vs. 248 (217; 284) mg/L, *p* < 0.001). The 25(OH)D_3_ levels remained suboptimal (24.5 (14.7; 29.5) ng/mL at the 3-month visit and 17.9 (12.4; 21.0) ng/mL at the 6-month visit, *p* = 0.052) despite recommended high-dose cholecalciferol supplementation. Patients also demonstrated an increase in 1,25(OH)_2_D_3_ levels (38 (31; 52) pg/mL at the 3-month visit and 49 (29; 59) pg/mL at the 6-month visit, *p* < 0.001) without a change in PTH or calcium levels during the follow-up. Conclusion: our results of a comprehensive laboratory evaluation of vitamin D status and mineral metabolism in patients undergoing bariatric surgery highlight the importance of improving current clinical guidelines, as well as careful monitoring and education of patients.

## 1. Introduction

Obesity is a well-known risk factor for vitamin D deficiency [1]. Several mechanisms are proposed to explain the high prevalence of low vitamin D levels in obese individuals: sequestration of vitamin D in adipose tissue and its decreased availability for the synthesis of active metabolite [2,3], volumetric dilution of ingested or cutaneously synthesized vitamin D in the fat mass [4,5], shade-seeking behavior [6] and reduced expression of 25-hydroxylase (CYP2R1) [7,8].

Vitamin D insufficiency is highly prevalent in patients undergoing bariatric surgery [9]. The prevalence of secondary hyperparathyroidism due to vitamin D deficiency is also relatively high in morbidly obese patients before bariatric surgery and increases continually after surgical treatment [10,11,12]. Some groups have shown a positive correlation between parathyroid hormone (PTH) and body mass index (BMI) [13,14] or duration of obesity [15]. However, several studies have also shown that PTH is suppressed at a relatively low concentration of 25(OH)D in obese patients [16,17,18]. The study by Barsin et al. demonstrated an increase in the cut-off point of 25(OH)D for PTH suppression following bariatric surgery [17]. In a work by Salazar et al., the threshold for a significant PTH increase was lower in patients who maintained obesity after bariatric surgery [18]. Furthermore, calcium–citrate clamping studies in obese subjects revealed a remarkably high sensitivity for calcium and a left-shifted relation between plasma calcium and PTH compared with the normal population [15]. These findings indicate altered regulatory mechanisms of calcium and vitamin D metabolism in obese patients and a role for bariatric surgery in regulating vitamin D metabolism.

Previously published data regarding active vitamin D metabolite (1,25(OH)_2_D) levels in obesity are conflicting. The two largest observational studies demonstrated significantly lower levels of 1,25(OH)_2_D in obese patients than in non-obese patients [19,20]; however, some of the studies, mostly smaller and older, showed opposite results [21,22,23,24,25]. This discrepancy is probably a consequence of the difference in methods used for 1,25(OH)_2_D measurement. In a study of post-bariatric patients, high-normal or elevated 1,25(OH)_2_D levels were reported against the background of secondary hyperparathyroidism [26,27]. Some studies also showed 1,25(OH)_2_D elevation after the surgery, with PTH levels being stable or returning to baseline after the initial decrease [28,29]. A lack of post-operative changes in 1,25(OH)_2_D has also been reported [30]. Another study evaluated the vitamin D metabolome in post-bariatric patients in the setting of calcifediol treatment and showed a decrease in 1,25(OH)_2_D levels and stable 24,25(OH)_2_D levels during follow-up [31].

Sporadic studies in which free 25(OH)D and vitamin D-binding protein (DBP) levels were investigated in comparison between obese vs. normal-weight subjects showed lower levels of free 25(OH)D and higher or equal levels of DBP in obesity [32,33,34]. The only study to date evaluating the changes in free 25(OH)D in patients undergoing bariatric surgery showed a correlation between serum total 25(OH)D and free 25(OH)D concentrations both before and after surgery and an increase in free 25(OH)D concentrations without changes in total 25(OH)D concentrations after surgery [35]. These data support the hypothesis that free 25(OH)D is more lipophilic than the protein-bound form and hence increases to a greater extent after weight loss.

Determining the dosage regimen for vitamin D supplementation in patients undergoing bariatric intervention remains challenging. According to the recent systematic review and meta-analysis, hypovitaminosis D was common after both restrictive and malabsorptive forms of bariatric procedures independently of the biochemical definitions, even with high-dose supplementation (defined as ≥2000 IU/day) [9]. Given that patients after malabsorptive types of surgery were characterized by more prominent alterations of mineral and vitamin D metabolism [11,36], decreased intestinal absorption of vitamin D and calcium seems to be the principal reason. The possible impact of residual obesity after surgery is also suggestive.

Overall, the studies carried out to date have a number of limitations, the major one being the assessment of a narrow spectrum of metabolites. In most studies, the determination of vitamin D was performed by immunological methods, which do not allow a separate study of the epimeric forms of vitamin D. The use of a mass spectrometry-based method allows for the separate determination of both classical metabolites of vitamin D (25(OH)D, 1,25(OH)_2_D, 24,25(OH)_2_D) and the epimeric form (3-epi-25(OH)D), and thus is essential to obtain a sufficient understanding of the pathogenesis of impaired vitamin D metabolism in these patients.

The aim of this study was to perform a comprehensive laboratory evaluation of vitamin D status and mineral metabolism in patients undergoing bariatric surgery, both before surgery and during postoperative follow-up for 6 months.

## 2. Materials and Methods

### 2.1. Study Population and Design

This was a prospective controlled cohort study that included 30 patients with morbid obesity referred for bariatric surgery treatment (one-anastomosis gastric bypass, OAGB) between February 2022 and July 2023. Patients were scheduled for a 6-month follow-up with laboratory examination at baseline, 3 and 6 months after bariatric surgical treatment. During the first 3 months, patients were prescribed therapy with cholecalciferol (50,000 IU per week in the form of an aqueous solution), with subsequent correction of therapy based on the results of the examination after 3 months of observation.

The control group included 30 healthy volunteers of similar age, sex and baseline 25(OH)D_3_. The exclusion criteria were the presence of granulomatous disease, malabsorption syndrome, or liver failure; decreased glomerular filtration rate (less than 60 mL/min per 1.73 m^2^); hypercalcemia for both groups; and calcium or vitamin D supplementation for 3 months prior to the study for the control group.

The laboratory assessment included vitamin D metabolites (25(OH)D_3_, 25(OH)D_2_, 1,25(OH)_2_D_3_, 3-epi-25(OH)D_3_, and 24,25(OH)_2_D_3_), PTH, DBP, free 25(OH)D, fibroblast growth factor 23 (FGF-23) and biochemical parameters (total calcium, albumin, phosphorus, creatinine and magnesium) in serum. The albumin-adjusted serum calcium levels were calculated using the formula: total plasma calcium (mmol/L) = measured total plasma calcium (mmol/L) + 0.02 × (40 − measured plasma albumin (g/L)) [37].

Serum samples were either transferred directly to the laboratory for biochemical analysis and PTH measurement or were stored at −80 °C, avoiding repeated freeze–thaw cycles for measurement of vitamin D metabolites, DBP, free 25(OH)D and FGF-23 at a later date. This study protocol was approved by the Ethics Committee of Endocrinology Research Centre, Moscow, Russia, on 24 February 2022 (abstract of record No. 4), and all participants signed an informed consent to participate in the study.

### 2.2. Laboratory Measurements

The levels of vitamin D metabolites in serum were determined by ultra-high performance liquid chromatography–tandem mass spectrometry (UPLC-MS/MS) using an Agilent 1290 Infinity II liquid chromatography system (Agilent Technologies, Santa Clara, CA, USA) equipped with a 4-channel Flexible pump, a multisampler, column thermostat, and an AB Sciex QTrap 5500 mass spectrometer (AB Sciex, Framingham, MA, USA) equipped with a TurboV Ion source (ESI) and a SelexION differential mobility separation device. An in-house developed UPLC-MS/MS method was used [38]. With this technique, the laboratory participates in the DEQAS quality assurance program (lab code 2388), and the results fall within the target range for the analysis of 25(OH)D and 1,25(OH)_2_D metabolites in human serum.

Chromatographic separation was carried out using an Acquity UPLC HSS PFP column (2.1 × 50 mm, 1.8 µm particle size, Waters), maintained at 40 °C. The three-component mobile phase was applied at a flow rate of 0.4 mL/min in gradient mode. Sample injection volume was 80 uL. Detection was performed in the positive ion mode. Ion source and differential mobility separation device parameters are shown in Table 1:

Nitrogen was used as the carrier gas. A carrier gas modifier and the resolution enhancement mode were not used. Registration of the components was carried out in scheduled multiple reaction monitoring (sMRM). All quantifiers, qualifiers and MRM parameters are shown in Table 2:

Sample preparation was performed using liquid–liquid extraction followed by solid-phase extraction. A 50 µL of the internal standard mixture solution (25(OH)D_3_-d_6_, 1,25(OH)_2_D_3_-d_6_, 3-epi-25(OH)D_3_-d_3_, 24,25(OH)_2_D_3_-d_6_ and D_3_-d_7_) was added to 300 µL of serum, vortexed and equilibrated for 5 min. Then, 900 µL of EtOAc was added, followed by extraction (15 min) and centrifugation (14,800 rpm, 6 min, 25 °C). The organic layer was isolated and dried using a vacuum centrifuge (40 °C, 1350 rpm, 5 mbar vacuum). Solid residue was reconstituted in a 4:6 methanol/water mixture (1 mL), centrifuged and loaded onto an Agilent Bond Elut C18 (50 mg, 1 mL) cartridge preconditioned with 0.5 mL of methanol and 1 mL of water. The cartridge was then washed with water followed by a 3:7 methanol/water mixture (1 mL of each), and analytes were eluted with 2 × 600 µL of methanol. The eluate was evaporated to dryness. A total of 115 µL of 1:1 methanol/water mixture was added to the residues; after 10 min of stirring in a shaker, samples were centrifuged (14,800 rpm, 10 min, 5 °C) and transferred to a 96-well plate.

Serum DBP, free 25(OH)D and FGF-23 levels were measured by enzyme-linked immunosorbent assay (ELISA) using commercial kits. The assay used for free 25(OH)D levels assessment (DIAsource, ImmunoAssays S.A., Ottignies-Louvain-la-Neuve, Belgium) has <6.2% intra- and inter-assay coefficient of variation (CV) at levels < 18.4 pg/mL. The assay used for DBP levels assessment (Assaypro, Saint Charles, MO, USA) has a 6.2% average intra-assay CV and a 9.9% average inter-assay CV. The assay used for FGF-23 levels assessment (Biomedica Medizinprodukte GmbH, Vienna, Austria) has a 12% average intra-assay CV and 10% average inter-assay CV at physiological range concentrations.

PTH levels were evaluated by electrochemiluminescence immunoassay (ELECSYS, Roche, Switzerland, Switzerland). Biochemical parameters of blood serum were assessed by ARCHITECT c8000 analyzer (Abbott, Chicago, IL, USA) using reagents from the same manufacturer according to standard methods.

### 2.3. Other Measurements

Clinical data were collected from patients’ medical records. Participants’ weight was measured in light indoor clothing with a medical scale to the nearest 100 g, and their height was measured with a wall-mounted stadiometer to the nearest centimeter. BMI was calculated as weight in kilograms divided by height in meters squared.

### 2.4. Statistical Analysis

Statistical analysis was performed using Statistica version 13.0 (StatSoft, OK, USA). Continuous variables were presented as median and interquartile range (IQR), and binary variables were summarized as numbers and percentages. The Mann–Whitney U-test and Fisher’s exact test were used for comparisons between the study group and the control group. Friedman ANOVA was performed to evaluate changes in the study group throughout the follow-up evaluation and pairwise comparisons using Wilcoxon test with adjustment for multiple comparisons (Bonferroni) were also made if the Friedman ANOVA was significant. Spearman rank correlation method was used to obtain correlation coefficients among indices. A *p*-value of less than 0.05 was considered statistically significant. When adjusting for multiple comparisons, a *p*-value greater than the significance threshold, but less than 0.05 was considered as a trend towards statistical significance.

## 3. Results

The general characteristics of the study group and the reference group are presented in Table 3. The groups were similar in terms of age, sex and baseline 25(OH)D_3_ status. The absolute majority of the participants in both groups presented with insufficient vitamin D levels, corresponding to vitamin D deficiency (defined as 25(OH)D_3_ below 20 ng/mL) in 2/3 of the participants. Eleven (37%) and 7 (23%) bariatric patients and healthy volunteers correspondingly had 25(OH)D_3_ levels below 10 ng/mL, indicating severe vitamin D deficiency. Only 3 patients (10%) from the study group reported irregular intake of vitamin D supplements (14,000 IU per week in two patients and 50,000 IU per week in one patient), and none of the patients reported intake of calcium supplements prior to enrollment.

Patients referred for bariatric surgery were diagnosed with several comorbidities of obesity: all patients had diabetes mellitus type 2 with suboptimal disease control on drug treatment (median HbA1c at the time of inclusion in the study equal 7.2 (6.2; 7.8) %, the median number of glucose-lowering drugs equal 3 (2; 4)) and dyslipidemia, and 28 patients (93%) had arterial hypertension.

At the baseline, patients with morbid obesity were characterized by significantly lower levels of active vitamin D metabolite 1,25(OH)_2_D_3_ (20 (10; 37) vs. 39 (33; 50) pg/mL, *p* < 0.001), lower serum calcium levels (2.24 (2.20; 2.32) vs. 2.31 (2.25; 2.35) mmol/L, *p* = 0.009) and magnesium levels (0.79 (0.72; 0.82) vs. 0.82 (0.78; 0.85) mmol/L, *p* = 0.043) with simultaneous similar 25(OH)D_3_ and PTH levels (*p* > 0.05) (Table 4). PTH levels were elevated in 6 patients (20%) from the study group and in 4 individuals (13%) from the control group (*p* = 0.731). DBP levels were higher in the study group (328 (288; 401) vs. 248 (217; 284) mg/L, *p* < 0.001); however, free 25(OH)D levels were similar between the groups (*p* = 0.129). The rest of the studied parameters also did not differ between the groups.

In bariatric patients, BMI was inversely correlated with serum calcium (rho = −0.370, *p* = 0.048) and directly correlated with PTH (rho = 0.480, *p* = 0.008), while in the control group, there was a positive correlation between BMI and serum calcium (rho = 0.436, *p* = 0.016) and no correlation between BMI and PTH (rho = 0.255, *p* = 0.174). Patients in the study group also had no correlation of 25(OH)D_3_ with 25(OH)D_3_/24,25(OH)_2_D_3_ and free 25(OH)D (*p* > 0.05), while healthy volunteers had a distinctive negative correlation of 25(OH)D_3_ with 25(OH)D_3_/24,25(OH)_2_D_3_ (rho = −0.583, *p* < 0.001) and a prominent positive correlation with free 25(OH)D (rho = 0.821, *p* < 0.001). 1,25(OH)_2_D_3_ was inversely correlated with PTH (rho = −0.410, *p* = 0.027) in the study group, while there was no such correlation in healthy controls (*p* > 0.05). PTH did not correlate with 25(OH)D_3_ levels in both groups (*p* > 0.05).

The results of the follow-up evaluation after bariatric surgery are presented in Table 5. Nine patients (30%) were lost to observation due to organizational issues. After the surgery, a median weight loss of 12.4 (9.8; 13.5) kg/m^2^ was observed by the end of the follow-up. Only 12 patients (40%) reported intake of vitamin D according to the recommendations during the first 3 months of the follow-up; 6 patients (20%) and 4 patients (13%) received 50,000 IU cholecalciferol per week for 8 weeks and 4 weeks, respectively, with subsequent treatment self-cancellation or dose reduction, and the rest of the patients reported irregular vitamin D intake in lower than prescribed dosage. Nineteen patients (90%) reported cholecalciferol intake (median 50,000 (14,750; 50,000) IU per week) between the 3 and 6 months follow-up: 1 patient—70,000 IU/week, 12 patients—42,000–50,000 IU/week, 5 patients—14,000–15,000 IU/week, 1 patient—2800 IU/week; of those who took 50,000 IU per week, two patients self-cancelled the treatment after 2 months, one patient reduced the dosage to 14,000 IU per week as was prescribed. Eighteen patients (60%) reported intake of calcium supplements after the surgery.

While levels of 25(OH)D_3_ showed a tendency to rise by the 3-month period and then to decline by the end of the follow-up period against the background of the therapy received, with concordant tendencies in 3-epi-25(OH)D_3_ levels, the changes did not reach statistical significance. Only 6 patients (24%) had sufficient 25(OH)D_3_ levels (above 30 ng/mL) at the 3-month visit, and none of the patients presented with sufficient levels at the 6-month visit. Nine (36%) and 13 (62%) patients remained vitamin D deficient (25(OH)D_3_ below 20 ng/mL) 3 and 6 months after surgery, respectively.

However, the levels of 1,25(OH)_2_D_3_ increased markedly during the follow-up (*p*_1–3_ < 0.001). Interestingly, the levels of PTH remained stable despite the abovementioned changes (*p* = 0.678). PTH remained elevated in 1 patient (4%) and in 5 patients (24%) at the 3-month and 6-month visits, respectively. We also observed a slight clinically non-significant decrease in creatinine levels (*p* = 0.002). The rest of the parameters were stable throughout the follow-up. Six months after surgery, still, no correlation between 25(OH)D_3_ and 25(OH)D_3_/24,25(OH)_2_D_3_ and free 25(OH)D was observed (*p* > 0.05), while the previously mentioned correlation between BMI and serum calcium, as well as the relationship between 1,25(OH)_2_D_3_ and PTH, were no longer observed (*p* > 0.05).

## 4. Discussion

One of the most important findings of our study is the quite unsatisfactory results of maintaining adequate vitamin D status after the bariatric surgery even with very high recommended doses, along with patients’ adherence to regular follow-up leaving much to be desired. Although excellent patient-reported supplement protocol adherence was shown in several studies [42,43], as well as an association between adhering to supplements and reporting a higher micronutrient intake with more favorable biochemistry [44], according to the systematic review and meta-analysis, the adherence of longitudinal studies to micronutrient supplementation recommendations after bariatric surgery did not exceed 20%, and in patients supplemented per guidelines, significant decreases in micronutrient levels were unfortunately still observed [45]. Lower pre-operative calcium and magnesium serum levels compared to controls are consistent with data from the literature on the high prevalence of micronutrient deficiencies in patients with severe obesity referred for bariatric surgery [46]. Altogether, this emphasizes the importance of coherent efforts on thorough regular clinical and laboratory monitoring of the patients undergoing bariatric surgery, proper patient education with a focus on dietary and lifestyle recommendations, as well as continuous work on the improvement of the current guidelines.

The recent consensus on vitamin D status and supplementation before and after bariatric surgery advised treating all patients after bariatric procedures with at least 2000 IU daily vitamin D supplementation, and preferring parenteral routes of vitamin D administration when available, in patients who undergo malabsorptive procedures [9]. However, our results, as well as some of the other studies using very high doses of vitamin D [47,48] provide a hint of a hardly attainable ideal of maintaining vitamin D target levels via the enteral route, especially in the real clinical practice setting, and might point towards more open-minded usage of parenteral preparations in this cohort. There is also emerging evidence that calcifediol is highly effective in the correction and maintenance of vitamin D status in bariatric patients, likely due to a higher rate of intestinal absorption (when compared with cholecalciferol) [49] and suppression of 25-hydroxylase in obesity [7,8], as very promising results were shown in recent studies [31,50].

Our results are generally consistent with the previous studies regarding the correlation of PTH and BMI [13,14], the prevalence of secondary hyperparathyroidism and low vitamin D levels [10,11,12], low baseline 1,25(OH)_2_D_3_ levels [19,20] and their increase during the follow-up [26,27,28,29], and stable levels of 24,25(OH)_2_D_3_ after the surgery [31]. The increase in 1,25(OH)_2_D_3_ levels without change in PTH levels during the follow-up is also supported by some of the previous papers [28,29]; however, the reason for this observation is unclear and might be attributed to the increase in substrate due to vitamin D supplementation or the ongoing multifactorial bone resorption [51]. Interestingly, serum calcium levels did not change despite the rise in 1,25(OH)_2_D_3_ levels, presumably due to malabsorption. The study by Herrera-Martínez et al. showed a decrease in 1,25(OH)_2_D_3_ after the surgical treatment; however, it was observed in the setting of calcifediol treatment [31].

Some of the studies showed different results with higher PTH and/or 1,25(OH)_2_D_3_ in bariatric surgery patients compared to control [20,22]. It should be noted that patients’ 25(OH)D_3_ levels in these studies were also lower, which might have an impact on the results obtained, as synthesis of the active metabolite is, on the one hand, might be dependent on the availability of the substrate, but on the other hand, is a priority in conditions of insufficient vitamin D content in order to implement its biological functions [52]. Our study group and control group were similar not only in terms of age and sex but also in basal 25(OH)D_3_ levels, which should be considered an important strength of our study.

A higher DBP in bariatric patients was consistent with previous data in obese patients [32,33] and might be explained by altered estrogen influence due to higher free estrogen levels [53] and proinflammation cytokine impact on DBP synthesis [54] of allelic variants [55]. While previous studies demonstrated lower free 25(OH)D levels in obesity [32,33,34], which was not observed in our study, it is noteworthy that total 25(OH)D levels were also lower in those cohorts. However, we observed a lack of correlation between 25(OH)D_3_ and free 25(OH)D, which is usually observed in normal individuals [56], indicating altered regulation of serum free 25(OH)D concentration in obesity. The lack of a standardized method of free 25(OH)D and DBP determination greatly complicates the unification of these data.

Our study, however, also had several limitations. One of them is the relatively small sample size. Second, the patients did not adhere to a strict follow-up schedule, which led to a discrepancy in the time intervals of the evaluation. Additionally, we did not evaluate calcium and phosphorus renal excretion, as well as laboratory parameters of bone metabolism, or body fat percentage, which might provide additional insights. This study was conducted on a homogenous Caucasian group of patients and in a specific geographical region; therefore, the results may not be easily translated to other regions with more UV and sunlight exposure as well as to other races and ethnicities with different skin types and genetic features. Also, in our study, patients’ dietary intake of vitamin D was not assessed. This was considered as not reasonable, since even in countries with mandatory vitamin D food fortification, vitamin D intake with food remains significantly lower than the recommended norm, according to large epidemiological studies [57], while in Russia vitamin D fortification is carried out only on a voluntary basis, and thus the percentage of fortified products on the market is substantially lower. Nevertheless, since the results of systematic reviews and meta-analyses indicate that fortification with vitamin D improves serum 25(OH)D concentrations [58,59], the extrapolation of our results to a population with a different approach to food fortification may also be controversial.

In conclusion, our data showed that patients referred for malabsorptive bariatric surgery had a high prevalence of vitamin D deficiency preoperatively and suboptimal 25(OH)D_3_ levels despite high-dose cholecalciferol supplementation. They also demonstrated lower baseline 1,25(OH)_2_D_3_ and higher DBP levels as compared to controls and an increase in 1,25(OH)_2_D_3_ levels without a change in PTH or calcium levels during the follow-up. The findings highlight the importance of improving current clinical guidelines, as well as careful monitoring and education of patients undergoing bariatric treatment.

## Figures and Tables

**Table 1 jcm-13-00007-t001:** Ion source and differential mobility separation device parameters.

Parameter	Description	Value
IS	Ion Spray Voltage	5500 V
GS1	Gas 1 Pressure	50 psi
GS2	Gas 2 Pressure	60 psi
Cur	Curtain Gas Pressure	28 psi
TEM	Source Heater Temperature	650 °C
SV	Separation Voltage	3800 V
COV	Compensation Voltage	6.8 V
OFS	Offset Voltage	−20 V
IHT	Interface Heater Temperature	150 °C

**Table 2 jcm-13-00007-t002:** MRM transitions used for detection of the vitamin D metabolites.

Analyte	Transition Type	Q1	Q3	CE, V	DP, V	CXP, V
1,25(OH)_2_D_3_	quantifier	399.3	135.1	28	89	16
	qualifier	399.3	381.3	19	89	14
24,25(OH)_2_D_3_	quantifier	417.3	399.3	13	66	15
	qualifier	417.3	381.3	15	66	14
25(OH)D_3_	quantifier	401.3	383.3	13	59	15
	qualifier	401.3	365.4	17	59	13
3-epi-25(OH)D_3_	quantifier	401.29	383.2	14	110	9
	qualifier	401.29	365.3	17	110	9
25(OH)D_2_	quantifier	413.3	355.3	15	110	7
	qualifier	413.3	395.3	13	110	7
1,25(OH)_2_D_3_-d_6_	IS	405.3	135.0	30	170	12
24,25(OH)_2_D_3_-d_6_	IS	423.3	387.5	16	150	7
25(OH)D_3_-d_6_	IS	407.4	389.3	12	120	11
3-epi-25(OH)D_3_-d_3_	IS	404.4	368.3	18	150	6

**Table 3 jcm-13-00007-t003:** Baseline general characteristics of the study group and the reference group.

Parameter	Study Group (n = 30)	Reference Group (n = 30)	*p*
Age, years	45.0 (39.0; 56.0)	42.5 (30.0; 59.0)	0.383
Sex, female/male	19(63%)/11(37%)	19(63%)/11(37%)	1.0
Baseline BMI, kg/m^2^	46.9 (40.5; 51.3)	25.6 (21.6; 28.3)	* <0.001
Baseline 25(OH)D_3_, ng/mL	11.9 (6.8; 22.2)	12.2 (10.2; 18.5)	0.744

Abbreviations: BMI, body mass index. * Significant difference in between-group comparison.

**Table 4 jcm-13-00007-t004:** Baseline laboratory assessment of the study group and the reference group.

Laboratory Parameter	Study Group (n = 30)	Reference Group (n = 30)	*p* (Mann–Whitney)	Reference Range
25(OH)D_3_, ng/mL	11.9 (6.8; 22.2)	12.2 (10.2; 18.5)	0.744	>30 ^a^
3-epi-25(OH)D_3_, ng/mL	1.1 (0.8; 2.4)	1.0 (0.5; 1.7)	0.194	Not available
1,25(OH)_2_D_3_, pg/mL	20 (10; 37)	39 (33; 50)	* <0.001	25–66 ^b^
24,25(OH)_2_D_3_, ng/mL	0.8 (0.5; 1.4)	0.8 (0.5; 1.5)	0.994	0.5–5.6 ^b^
25(OH)D_3_/24,25(OH)_2_D_3_	15.3 (12.1; 20.9)	16.9 (10.8; 25.9)	0.581	7–23 ^b^
Free 25(OH)D, pg/mL	5.2 (4.1; 6.5)	4.2 (3.8; 5.5)	0.129	2.4–35 ^c^
DBP, mg/L	328 (288; 401)	248 (217; 284)	* <0.001	176–623 ^c^
FGF-23, pmol/L	1.1 (0.7; 1.4)	0.9 (0.5; 1.2)	0.319	Not available
Albumin-adjusted calcium, mmol/L	2.24 (2.20; 2.32)	2.31 (2.25; 2.35)	* 0.009	2.15–2.55 ^c^
Phosphorus, mmol/L	1.15 (1.09; 1.31)	1.16 (1.00; 1.30)	0.589	0.74–1.52 ^c^
PTH, pg/mL	41.4 (31.5; 57.8)	40.7 (31.1; 52.2)	0.912	15–65 ^c^
Creatinine, µmol/L	67.7 (61.6; 77.7)	69.9 (65.1; 79.2)	0.352	63–110 (male) 50–98 (female) ^c^
Magnesium, mmol/L	0.79 (0.72; 0.82)	0.82 (0.78; 0.85)	* 0.043	0.7–1.05 ^c^

Abbreviations: PTH, parathyroid hormone; DBP, vitamin D-binding protein; FGF-23, fibroblast growth factor 23. ^a^ Reference range is given for total of 25(OH)D according to the clinical guidelines [1,39]; the 25(OH)D_2_ fraction is negligible (<0.5 ng/mL in absolute values) for the purposes of this study. ^b^ Reference ranges are given according to the data in the literature [40,41]. ^c^ Reference ranges are specified according to kit manufacturers’ recommendations. * Significant difference in between-group comparison.

**Table 5 jcm-13-00007-t005:** Results of the follow-up evaluation in the study group.

Parameter	Baseline (n = 30)	3 Months (n = 25)	6 Months (n = 21)	*p* (Friedman ANOVA)	*p* (Wilcoxon) **	Reference Range
Actual visit time, days from surgery	-	101 (90; 119)	206 (192; 236)	-	-	-
BMI, kg/m^2^	46.9 (40.5; 51.3)	37.1 (30.9; 41.6)	35.8 (30.5; 38.9)	* <0.001	*p*_1–2_ < 0.001 *p*_2–3_ < 0.001 *p*_1–3_ < 0.001	Not available
25(OH)D_3_, ng/mL	11.9 (6.8; 22.2)	24.5 (14.7; 29.5)	17.9 (12.4; 21.0)	0.052	*p*_1–2_ = 0.023 *p*_2–3_ = 0.030 *p*_1–3_ = 0.711	30–100 ^a^
3-epi-25(OH)D_3_, ng/mL	1.1 (0.8; 2.4)	3.3 (1.9; 4.7)	1.2 (0.9; 2.1)	0.056	*p*_1–2_ = 0.053 *p*_2–3_ = 0.009 *p*_1–3_ = 0.286	Not available
1,25(OH)_2_D_3_, pg/mL	20 (10; 37)	38 (31; 52)	49 (29; 59)	* <0.001	*p*_1–2_ = 0.023 *p*_2–3_ = 0.594 *p*_1–3_ < 0.001	25–66 ^b^
24,25(OH)_2_D_3_, ng/mL	0.8 (0.5; 1.4)	1.5 (0.9; 2.1)	1.1 (0.9; 1.6)	0.355	-	0.5–5.6 ^b^
25(OH)D_3_/24,25(OH)_2_D_3_	15.3 (12.1; 20.9)	16.5 (11.9; 22.3)	14.2 (12.4; 16.2)	0.458	-	7–23 ^b^
Free 25(OH)D, pg/mL	5.2 (4.1; 6.5)	6.6 (5.5; 7.9)	5.3 (3.8; 5.8)	0.838	-	2.4–35 ^c^
DBP, mg/L	328 (288; 401)	342 (314; 385)	370 (329; 452)	0.327	-	176–623 ^c^
FGF-23, pmol/L	1.1 (0.7; 1.4)	1.3 (1.0; 1.7)	1.0 (0.7; 1.9)	0.589	-	Not available
Albumin-adjusted calcium, mmol/L	2.24 (2.20; 2.32)	2.31 (2.22; 2.35)	2.24 (2.20; 2.28)	0.080	-	2.15–2.55 ^c^
Phosphorus, mmol/L	1.15 (1.09; 1.31)	1.22 (1.08; 1.29)	1.27 (1.20; 1.38)	0.299	-	0.74–1.52 ^c^
PTH, pg/mL	41.4 (31.5; 57.8)	42.5 (34.3; 58.1)	42.8 (34.1; 65.1)	0.678	-	15–65 ^c^
Creatinine, µmol/L	67.7 (61.6; 77.7)	63.4 (59.2; 71.0)	62.1 (56.4; 65.9)	* 0.002	*p*_1–2_ = 0.492 *p*_2–3_ = 0.005 *p*_1–3_ < 0.001	63–110 (male) 50–98 (female) ^c^
Magnesium, mmol/L	0.79 (0.72; 0.82)	0.76 (0.73; 0.84)	0.76 (0.72; 0.82)	0.790	-	0.7–1.05 ^c^

Abbreviations: BMI, body mass index; PTH, parathyroid hormone; DBP, vitamin D-binding protein; FGF-23, fibroblast growth factor 23. ^a^ Reference range is given for total of 25(OH)D according to the clinical guidelines [1,39]; the 25(OH)D_2_ fraction is negligible (<0.5 ng/mL in absolute values) for the purposes of this study. ^b^ Reference ranges are given according to the data in the literature [40,41]. ^c^ Reference ranges are specified according to kit manufacturers’ recommendations. * Significant difference in between-group comparison. ** *p*_1–2_—baseline–3 months comparison; *p*_2–3_—3 months–6 months comparison; *p*_1–3_—baseline–6 months comparison.

## Data Availability

The datasets generated during and/or analyzed during the current study are available from the corresponding author upon reasonable request.

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
