# Peer review of "Dynamic Evaluation of Vitamin D Metabolism in Post-Bariatric Patients"

_jcm, 2023, doi:10.3390/jcm13010007_

Round 1

Reviewer 1 Report

Comments and Suggestions for Authors

This article is informative.

Laboratory evaluation of vitamin D status in patients underwent bariatric surgery, before and during postoperative follow-up, was performed.

I consider the topic relevant in the field. The article emphasised that  in bariatric surgery monitoring and education of patients  is important.

New researches in this field would be welcome in future.

References are appropriate.

Comments on the Quality of English Language

Minor editing of English language required.

Author Response

Dear Reviewer

Many thanks for taking the time to review our paper.

Reviewer 2 Report

Comments and Suggestions for Authors

The paper “Dynamic evaluation of vitamin D metabolism in post-bariatric patients” is adequate for the Journal of Clinical Medicine and within the scope of the Journal.

Some minor concerns should be addressed.

If the patients were enrolled in the period between February 2022 and July 2023 (line 98) than how is it possible to have 6-month follow up from July 2023 having on mind that the paper is being reviewed in November 2023 (less than five months from July 2023)?

Lines 121-122: The levels of vitamin D metabolites in serum were determined by ultra-high  performance liquid chromatography-tandem mass spectrometry (UPLC-MS/MS) using  an in-house developed method, described earlier.

Please provide more details about sample preparation before UPLC-MS/MS is applied? Was any preconcentration, extraction etc applied? Also please provide some data regarding the UPLC-MS/MS although the method has been described earlier.

Lines 300-304: Our study, however, also had several limitations. One of them is the relatively small sample size. Second, the patients did not adhere to a strict follow-up schedule, which led  to a discrepancy in the time intervals of the evaluation. Additionally, we did not evaluate calcium and phosphorus renal excretion, as well as laboratory parameters of bone metabolism, or body fat percentage, which might provide additional insights.

I would also add that the study is conducted on homogenous Caucasian group of patients presumably from Russia (presumably from the same region – not clearly defined in the paper), thus the results may not be easily extrapolated to other geographical regions with more UV and sunlight as well to other races and ethnicities with different skin color and melanin content.

Author Response

Dear Reviewer

Many thanks for taking the time to review our paper. We deeply appreciate your comments and recommendations for improvements, please find below our answers. 

The paper “Dynamic evaluation of vitamin D metabolism in post-bariatric patients” is adequate for the Journal of Clinical Medicine and within the scope of the Journal.

Some minor concerns should be addressed.

If the patients were enrolled in the period between February 2022 and July 2023 (line 98) than how is it possible to have 6-month follow up from July 2023 having on mind that the paper is being reviewed in November 2023 (less than five months from July 2023)?

The last patient from the cohort which was included in July 2023, was lost to observation as well as some other patients, as stated in line 243.

Lines 121-122: The levels of vitamin D metabolites in serum were determined by ultra-high performance liquid chromatography-tandem mass spectrometry (UPLC-MS/MS) using an in-house developed method, described earlier.

Please provide more details about sample preparation before UPLC-MS/MS is applied? Was any preconcentration, extraction etc applied? Also please provide some data regarding the UPLC-MS/MS although the method has been described earlier.

The description of the method in the article has been expanded. We also updated the reference regarding the previous description of the method, and included publication with detailed explanation of the procedure.

Lines 300-304: Our study, however, also had several limitations. One of them is the relatively small sample size. Second, the patients did not adhere to a strict follow-up schedule, which led to a discrepancy in the time intervals of the evaluation. Additionally, we did not evaluate calcium and phosphorus renal excretion, as well as laboratory parameters of bone metabolism, or body fat percentage, which might provide additional insights.

I would also add that the study is conducted on homogenous Caucasian group of patients presumably from Russia (presumably from the same region – not clearly defined in the paper), thus the results may not be easily extrapolated to other geographical regions with more UV and sunlight as well to other races and ethnicities with different skin color and melanin content.

The discussion of the limitations of the study has been expanded according to the suggestion.

Reviewer 3 Report

Comments and Suggestions for Authors

Dear Authors,

I am grateful for the opportunity to review the manuscript entitled "Dynamic Evaluation of Vitamin D Metabolism in Post-Bariatric Patients" authored by Povaliaeva et al. The manuscript offers a noteworthy exploration of vitamin D, its metabolites, and other molecules in patients post-bariatric surgery. Overall, the presentation of the article is commendable, and it furnishes pertinent information. I would like to offer two minor suggestions for consideration:

  1. In reference to the Spearman's correlation coefficient values, the standard notation employed is "rho" or "rs". May I suggest the authors consider revising the current format to align with this convention?

  2. An expanded discussion regarding the patients' dietary intake of vitamin D would be beneficial. Such information could provide a more comprehensive understanding of the study results. Additionally, it could illuminate the extent to which dietary factors might serve as confounding variables in this study.

  3. Although it is described that the study was approved, its approval number is not added. Please add

I look forward to the authors' response to these suggestions and appreciate their contribution to the field.

Author Response

Dear Reviewer

Many thanks for taking the time to review our paper. We deeply appreciate your comments and recommendations for improvements, please find below our answers. 

  1. The format of the correlation coefficient has been revised.
  2. In our study, the patients' dietary intake of vitamin D was not assessed. This was considered as not reasonable, since even in countries with mandatory vitamin D food fortification, vitamin D intake with food remains significantly lower than the recommended norm, according to large epidemiological studies [doi: 10.17226/13050], while in Russia, vitamin D fortification is carried out only on a voluntary basis, thus the percentage of fortified products on the market is substantially lower.

However, this aspect is indeed important to discuss in the limitations. Since the results of systematic reviews and meta-analyses nevertheless indicate that fortification with vitamin D improves serum 25(OH)D concentrations [doi: 10.1017/S0007114521002816, 10.1093/jn/nxab180], the extrapolation of our results to a population with a different approach to food fortification may be questionable.

The corresponding discussion was added to the manuscript.

  1. The approval number of the study has been added (line 125).